# A Preliminary Study on the Relationship between Gastric Lesions and Anti-Inflammatory Drug Usage in Heavy Pigs

**DOI:** 10.3390/vetsci10090551

**Published:** 2023-09-02

**Authors:** Sergio Ghidini, Federico Scali, Claudia Romeo, Federica Guadagno, Antonio Marco Maisano, Silvio De Luca, Maria Olga Varrà, Mauro Conter, Adriana Ianieri, Emanuela Zanardi, Giovanni Loris Alborali

**Affiliations:** 1Department of Food and Drug, University of Parma, Strada del Taglio 10, 43126 Parma, Italy; sergio.ghidini@unipr.it (S.G.); silvio.deluca@unipr.it (S.D.L.); mariaolga.varra@unipr.it (M.O.V.); adriana.ianieri@unipr.it (A.I.); emanuela.zanardi@unipr.it (E.Z.); 2Istituto Zooprofilattico Sperimentale della Lombardia e dell’Emilia Romagna “Bruno Ubertini”, 25124 Brescia, Italy; federico.scali@izsler.it (F.S.); claudiarosa.romeo@izsler.it (C.R.); federica.guadagno@izsler.it (F.G.); antoniomarco.maisano@izsler.it (A.M.M.); giovanni.alborali@izsler.it (G.L.A.); 3Department of Veterinary Medicine Sciences, University of Parma, Strada del Taglio 10, 43126 Parma, Italy

**Keywords:** animal welfare, gastric lesions, anti-inflammatory drugs, abattoir, monitoring schemes

## Abstract

**Simple Summary:**

Gastric lesions are a common condition in pigs, representing a critical issue in the pig industry that can cause heavy losses for farmers in terms of reduced growing performance and mortality. In heavy pig production, which is typical of Italy and is characterized by a long fattening period, the prevalence of such lesions at the abattoir has not been widely studied. Moreover, the impact of steroidal and non-steroidal anti-inflammatory drug (NSAIDs) usage on the occurrence of such lesions in heavy pigs needs to be better investigated. In this study, the prevalence of gastric lesions in heavy pigs in two large slaughterhouses was investigated. Furthermore, the association between the usage of anti-inflammatory drugs and gastric lesions was explored in a subset of farms. Most of the animals had mild or severe ulcers, and a relationship with the usage of NSAIDs was found. The results of this study suggest that gastric lesions are an important issue for heavy pig production and that non-selective NSAIDs should be used with caution on farms where the risk factors for gastric ulcers are common.

**Abstract:**

Gastric lesions in pigs cause welfare and economic losses. Their prevalence in heavy pigs reared for premium products (e.g., Parma ham) requires further investigation. Stress, nutrition, and farm management are known risk factors, but the effects of steroidal and non-steroidal anti-inflammatory drugs (NSAIDs) are largely unknown. The aim of this study was to evaluate the prevalence of gastric lesions in Italian heavy pigs and their possible association with the use of anti-inflammatory drugs. A total of 9371 pig stomachs from 76 farms were evaluated. Among these, 20.3% showed no lesions, while 30.7%, 42.1%, and 6.8% were scored 1, 2 and 3, respectively. A tendency for an inverse relationship with farm size emerged. The use of steroids and NSAIDs was estimated by calculating a treatment incidence per 1000 (TI_1000_) in a subset of 36 farms. At least one prescription for NSAIDs and/or steroids was found in 80.6% of the farms (55.6% used NSAIDs and 63.9% used steroids). Median TI_1000_ was 0.07 (range: 0–30.1) and 0.18 (range: 0–6.2) for NSAIDs and steroids, respectively. Gastric scores were positively associated with NSAID use, but not with steroid use. Although the role of these drugs in gastric lesions needs to be further clarified, these findings suggest a cautious use of non-selective NSAIDs.

## 1. Introduction

Gastric lesions, particularly ulcers, are a major concern in the pig industry due to their impact on animal health, welfare, and performance. Severe gastric ulcers can lead to high mortality, accounting for up to 27% of total mortality in grower-finisher units [1]. Among other consequences, gastric ulcers can also affect the welfare of pigs, resulting in behavioural changes during the finishing stage, with affected animals spending more time standing, walking and changing their postures [2]. Gastric ulcers are usually examined at slaughter, and their prevalence in intensively reared pigs varies widely between studies, from less than 5% to 55% [3,4,5].

The development of gastric lesions involves several stages. It starts with an initial hyperkeratosis, which is a protective response of the mucosa to various noxae, followed by the onset of erosions. Finally, in the presence of persistent stimuli, deep ulcerations occur, which may be accompanied by haemorrhage or, in the case of healing processes, eschar formation [6]. Several factors are involved in the development of gastric lesions [7]. Diet is reported to be the main cause of such lesions [8], with the physical structure of the feed being the most important risk factor [9]. Fine particle size and pelleting, together with fasting and irregular feeding, significantly increase the prevalence and severity of gastric ulcers [10,11,12]. Other elements have also been reported as co-factors in the development of the disease, such as housing systems, feed withdrawal, sex, and genetics [13,14]. A recent study investigating the occurrence of gastric ulcers in piglets found a positive association between low birth weight and the frequency of gastric lesions in weaners. This result may help to explain differences in the development of gastric lesions in pigs exposed to similar environmental, management and feeding conditions [15].

In humans and other mammals, the development of gastric lesions has also been associated with the administration of anti-inflammatory drugs [16]. Specifically, the mechanism of gastric ulceration in humans is mediated by cyclooxygenase (COX), an enzyme that catalyses the conversion of arachidonic acid to prostaglandins. COX has two different isoforms, COX-1 and COX-2. COX-1 is normally expressed and present in healthy tissues, while COX-2 is mostly induced by inflammatory mediators when inflammation occurs. The anti-inflammatory effects of NSAIDs such as meloxicam and nimesulide are mainly due to the inhibition of the COX-2 enzyme, whereas inhibition of COX-1 causes gastrointestinal toxicity [17]. To date, only a few studies have investigated the efficacy of these drugs in pigs housed under different conditions and treatment regimens [18]. Most of these studies have focused on their efficacy in relieving pain in piglet castration and their anti-inflammatory potential in experimental models of inflammation [18,19,20]. The use of NSAIDs and their effects on the development of gastric lesions in fattening pigs have been little studied. Furthermore, information on the use of anti-inflammatory drugs in pig farms is scarce.

Anti-inflammatory drugs may also be useful in reducing antimicrobial use (AMU). For example, in dairy calves, the use of NSAIDs in combination with supportive therapy appears to be much more common in herds with low AMU [21]. In pig farming, NSAIDs are considered one of the most feasible alternatives to antimicrobials, although not one of the most effective [22]. Furthermore, the treatment of inflammation is important in the context of animal welfare, so the effects of anti-inflammatory use on pig health should be further investigated [23].

The aim of this study was to assess the prevalence of gastric lesions and the use of anti-inflammatory drugs in heavy pigs, while also investigating any potential links between these factors within the peculiar context of Italian heavy pigs (slaughtered at 160/170 kg body weight) intended for use in Protected Designation of Origin (PDO) products.

## 2. Materials and Methods

### 2.1. Pig Breeding and Feeding Information

For the production of Italian heavy pigs intended for PDO production, breed requirements are reported in the PDO specifications. In particular, as pure breeds, only individuals from the Italian Large White and Italian Landrace breeds can be used. Crosses with the Italian Duroc breed are also permitted. Other breeds can be used as long as they are compatible with the Italian Herd Book. Regarding the feeding of pigs, the presence of dry matter from grains during the fattening phase may not be lower than 55% of the total, both in early (from 90 to 130 kg BW) and late (from 130 to 170 kg BW) finishing periods. The PDO specifications report all types and concentrations of feed that can be used during the fattening period [24].

### 2.2. Sample Collection and Stomach Classification

Sampling was performed from October to December 2019 in two large slaughterhouses located in the Lombardy region (Northern Italy). The two abattoirs had a weekly output of about 15,000 pigs, slaughtering around 960,000 pigs per year. The abattoirs were selected for the following reasons: slaughtering of heavy pigs only, previous collaborations with our research group, and availability of dedicated space for stomach assessment.

The examined batches were randomly selected from samples made available at the slaughterhouses, without any prior information on their farms of origin. Within each batch, at least 70% of pigs were randomly selected and examined for gastric ulcers. Stomachs were categorised according to Robertson et al. [8], as follows: 0 (absence), 1 (hyperkeratosis), 2 (erosion or mild ulcer) and 3 (severe ulcer) (Figure 1). 

The assessment of gastric lesions was performed by two trained veterinarians after stomach washing around 45 minutes after slaughter, which occurred on both premises by head-only electrical stunning and bleeding of the animals after severing of the common brachiocephalic trunk in the neck. The two veterinarians were trained at the beginning of the study in order to reach a good level of agreement between observers. The training consisted of 6 full-day scoring sessions performed at both slaughterhouses (3 days at each abattoir) under the supervision of S.G. In particular, during the training sessions, the mucosa of the *pars oesophagea* of at least 100 stomachs per session was first assessed by S.G. and then by both trainees. In case of disagreement, uncertain evaluations were further discussed among all three assessors until an agreement was reached.

Farm-level data on the use of steroidal and nonsteroidal anti-inflammatory drugs (NSAIDs) during 2019 were collected via the Italian Electronic Prescription System [25]. At that time, however, the system was still in a refinement phase and data on anti-inflammatory usage were found to be available only for a limited number of the sampled farms. Therefore, only gastric scores of pigs provided by farms with complete data on 2019 usage were retained for further analysis. Dexamethasone was the only prescribed steroid. Paracetamol consumption was not considered in the study due to some missing data from the prescriptions of medicated feed (i.e., the concentration of the drug within the feed was not always present). Data concerning the number of reared pigs in 2019 and the type of production were collected using the Italian Veterinary Database [26]. 

### 2.3. Use of Anti-Inflammatory Drugs 

The consumption of NSAIDs and dexamethasone at farm level was estimated by calculating a treatment index per 1000 (TI_1000_), an indicator comparable to that used for the assessment of antimicrobial use in humans and animals [27]. A defined daily dose animal for Italy (DDDAit) was established for each prescribed medicine, taking into account the amount of active ingredient (in mg) to be administered per kg of live weight per day (mg/kg/d), as stated in the summary of the product characteristics. If the dosage was expressed as a range, the average amount was adopted (e.g., 8 to 12 mg/kg/d, DDDAit = 10). The TI_1000_ was calculated according to the following formula (modified from [27,28]):$${TI}_{1000}=\frac{active\;ingredient\;used\;(mg)\;per\;farm\;in\;2019}{DDDAit\;(mg/kg/d)\;\times \;animal\;at\;risk\;\times \;weight\;at\;risk\left(kg\right)\;\times \;days\;at\;risk}\times 1000$$

The number of pigs reared during 2019 was used as the “animal at risk” value and “weight at risk” and “days at risk” were estimated at 100 kg and 180 days [29], respectively. The TI_1000_ is a standardised unit of measurement that can be interpreted as the number of pigs treated on a given day per 1000 pigs housed on a farm.

### 2.4. Statistical Analysis

To analyse variation in gastric ulcer scores at the farm level, we first derived a single farm-level score by summing, within each farm, the weighed proportions of stomachs showing a given score, as follows:$$\sum_{i=1}^{n}s_{i}\times p_{i}$$
where *s_i_* is the *score_i_* and *p_i_* is the proportion of stomachs with *score_i_.*

Firstly, we explored the relationship between farm-level scores (*n* = 76), type of operation (farrow-to-finish or fattening) and farm size (average number of pigs housed in 2019) through a linear model. The dependent variable was power-transformed (x^2^) to achieve normality of residuals (Shapiro–Wilk test: W = 0.98; *p* = 0.34). In a subset of 36 fattening farms for which data on anti-inflammatory drugs usage was available, we analysed variation in the power-transformed farm-level scores through another linear model, using NSAID TI_1000_ and dexamethasone TI_1000_ as explanatory variables and farm size as a covariate. We also tested for any correlation between anti-inflammatory drug consumption (expressed as TI_1000_) and farm size through Spearman’s rank correlation. All the analysis was carried out using SAS/STAT 9.4 software (Copyright © 2011, SAS Institute Inc., Cary, NC, USA).

## 3. Results

The stomachs of 9371 pigs from 76 different farms were examined and scored for gastric ulcers (range of examined stomachs/farm: 91–546). Of these, 65 farms were fattening units, while 11 were farrow-to-finish. The median farm size was 2561 housed pigs (range: 211–16,400). Considering the whole dataset, 1902 stomachs (20.3%) did not present any lesions, 2875 (30.7%) were scored 1, 3954 (42.1%) were scored 2 and 640 (6.8%) were scored 3. The mean proportions of stomachs by scores observed at the farm level are shown in Figure 2. The median farm-level score over the whole sample was 1.42 (range: 0.10–2.25; Figure 3), and did not differ depending on the type of operation (F_1, 73_ = 1.45; *p* = 0.23). Although not significant (F_1, 73_ = 3.89; *p* = 0.052), there was a tendency for an inverse relationship between the score and farm size, with bigger farms showing lower scores.

At least one prescription of NSAIDs and/or dexamethasone was found on 29 farms, corresponding to 80.6% of the subset of 36 fattening farms with complete data on anti-inflammatory drug usage in 2019. The median TI_1000_ of overall consumption was 0.45 (range: 0–31.6). NSAIDs were used on 20 out of 36 farms (55.6%) with a median TI_1000_ of 0.07 (range: 0–30.1), while dexamethasone was used on 23 out of 36 farms (63.9%), with a median TI_1000_ of 0.18 (range: 0–6.2). Roughly 90% of the overall usage of anti-inflammatory drugs was attributed to NSAIDs, primarily because of a limited group of farms characterized by elevated NSAID consumption. Globally, oral products accounted for 99.5% of NSAIDs usage, while dexamethasone was only administered in injectable form. Detailed data on use of anti-inflammatory drugs are reported in Table 1.

Neither NSAID nor dexamethasone consumption were correlated with farm size (Spearman’s rho = 0.09 and 0.12; *p* = 0.61 and 0.49). The median farm-level ulcer score (computed for the subset of 36 fattening farms with complete information regarding the usage of anti-inflammatory drugs) was 1.42, the same as in the larger sample, with values ranging from 0.46 to 1.97. Variation in farm-level scores was positively associated with NSAID TI_1000_ (parameter estimate ± SE: 0.032 ± 0.015; F_1, 32_ = 4.43; *p* = 0.043), whereas dexamethasone consumption had no significant relationship to scores (F_1, 32_ = 1.93; *p* = 0.17). Farm size was inversely associated with the score, with bigger farms showing lower scores (parameter estimate ± SE: −0.0001 ± 0.00004; F_1, 32_ = 6.33; *p* = 0.017).

## 4. Discussion

Over half of the stomachs examined in this study displayed mild (score 2) or severe (score 3) lesions. Overall, this percentage is consistent with what was reported in a previous study on Italian heavy pigs [30], but considerably higher than what has been found in other studies (20–26%) [4,31]. Large differences in the prevalence of these lesions have also been reported in porker or baconer pigs, which are slaughtered at 90–110 kg [32]. Table 2 presents a summary of the occurrence and severity of swine gastric lesions as documented in published literature, along with a comparison to the findings of our study. For example, Guise et al. [33] reported a prevalence of mild and severe ulcers in pigs slaughtered in the United Kingdom of 9.5% and 13.4%, respectively, while Cybulski et al. [3] in Poland reported a prevalence of severe ulcers in 54.9% of the pigs. Such differences may reflect the diverse conditions under which pigs are reared and thus the presence of different risk factors. As it is a widespread practice in Italy [34,35,36], it can be assumed that the majority of farms did not have straw as bedding or did not have the “optimal” material reported by the Commission Recommendation (EU) 2016/336 [37] even if information on environment enrichment was not collected in the present study. Under these conditions, pigs tend to develop more gastric ulcers than those reared with permanent access to straw [38]. Indeed, access to enrichment materials such as straw seems to be a valid approach to reducing the incidence of these lesions [39]. Straw can increase the consistency of the stomach contents after ingestion, reducing the fluidity of the contents, which is a critical ulcerogenic factor [40].

While gastric scores were not related to the type of operation (fattening or farrow-to-finish), an inverse relationship was found to farm size. This latter result is in contrast to a previous study on porkers in which pigs from larger farms showed more gastric lesions than those from smaller farms [39]. In that study, it was suggested that other conditions causing inappetence or anorexia, such as respiratory or reproductive diseases, may have influenced the onset of gastric lesions [39]. A positive correlation between lung, pleura or liver lesions and gastric lesions has also been reported in heavy pigs [4]. As respiratory diseases are more frequent and severe during the winter months, this may also explain the increased risk of mortality due to severe ulcers in pigs during this period [49]. Regarding liver lesions, histamine release due to parasitic infection may also predispose the pigs to gastric lesions [7]. However, in addition to concurrent pathological conditions, other factors may influence the occurrence of gastric lesions, such as the housing system (e.g., the type of floor) [13], the availability of enrichment materials [39,40], and the diet regimen [28]. 

Quantitative data on the use of anti-inflammatory drugs in farm animals are generally scarce. A Belgian study [50] on white veal calves was carried out with a metric similar to our study, calculating a TI using a DDD-based approach. The average use of anti-inflammatories found in our study was slightly higher (5.9 vs. 5.1), as was the use of dexamethasone (0.70 vs. 0.45), with a wider range of consumption in pig farms than in veal farms. However, comparisons between the two studies may have limitations due to the various species involved and possible differences in dosage.

Among the potential risk factors for gastric lesions in pigs, the use of NSAIDs has rarely been investigated, although there is some evidence that these drugs can effectively induce ulcers in experimentally treated pigs [51,52]. To the best of our knowledge, no study has quantitatively investigated the association between anti-inflammatory drugs and gastric lesions. Gottardo et al. [4] have assessed the use of NSAIDs from a qualitative perspective (use/non-use), but no strong evidence of association with gastric lesions was found. Our data revealed that anti-inflammatory drugs were commonly used but with wide differences among farms. Although NSAIDs were administered less frequently than dexamethasone (55% versus 64% of the farms) and their median of consumption was lower (0.07 versus 0.18), they accounted for almost 90% of the total anti-inflammatory drug usage. Considering that NSAIDs were administered orally (group treatments) and dexamethasone only by injection (individual treatments), this result may potentially be explained either by a small cluster of farms characterized by high NSAID consumption or by the different drug administration route. Acetylsalicylic acid and sodium salicylate accounted for almost all the NSAIDs consumed. Considering the weak but significant relationship between NSAID usage and gastric lesions, the frequent administration of those two non-selective NSAIDs could at least partially explain the high frequency of severe lesions found in this study. However, in the specific context of gastric ulcer development, it is crucial to consider the potential contrast between the ulcerogenic properties of acetylsalicylic acid and sodium salicylate. Indeed, research studies have suggested that salicylic acid or its sodium derivative might comparatively cause less severe harm to the gastric mucosa, possibly due to the lack of COX activity and prostaglandin synthesis inhibition [53,54]. Nevertheless, the relationship between the utilization of NSAIDs and the occurrence of gastric lesions confirms that if gastric ulcer issues are suspected, the use of non-selective NSAIDs should be approached with caution. In such cases, COX-2 inhibitors (like meloxicam) should be given more consideration, as they are frequently and successfully used for treating various conditions in other species, for example, in the case of degenerative joint diseases and colic in horses [55]. However, given the low usage of selective NSAIDs in the involved farms, the role of these drugs should be investigated in further studies. 

Two important limitations of this study were the small number of farms involved and the lack of information on other risk factors for gastric ulcers in these farms. The availability of complete data on the use of anti-inflammatory drugs was limited at the time of the study, which reduced the number of eligible farms. Moreover, the occurrence of gastric lesions can be influenced by several risk factors, of which dietary factors may play a major role. Feed characteristics (e.g., that expanded maize is more ulcerogenic than raw maize) [56], processing characteristics (e.g., that pelleted feed is more ulcerogenic than non-pelleted feed) [57,58], straw provision [30], and access to grass silage [59] are all critical determinants of both occurrence and severity of gastric lesions. For these reasons, the results of this study should be considered preliminary. It cannot be ruled out that other factors influenced the prevalence of gastric ulcers and that the relationship with NSAIDs is spurious. However, with all its limitations, the results of this study imply that on farms dealing with gastric ulcer issues, a careful evaluation of NSAID administration, particularly a reduction in the use of non-selective agents, might be warranted.

Finally, even if the potential adverse effects on the stomach could be possible, the use of NSAIDs should not be discouraged given their importance in reducing pain and inflammation in respiratory disease and lameness, which are two of the most important diseases affecting finishers [18]. In addition, correct use of NSAIDs may reduce the need for antimicrobial treatments.

## 5. Conclusions

In this study, over 50% of the pigs examined had either mild or severe gastric lesions, confirming the importance of this problem in swine production. Anti-inflammatory drugs were commonly used, but with wide differences among farms. Acetylsalicylic acid and sodium salicylate were the most common drugs in terms of total consumption. A significant relationship was found between the use of NSAIDs and the presence of gastric lesions. However, such results should be considered as preliminary. Further studies are needed to confirm that association, taking into account other known risk factors such as diet or the availability of enrichment materials.

The use of anti-inflammatory drugs in pigs should not be discouraged, as they can improve animal welfare and reduce the overall usage of antimicrobials. If gastric ulcer issues are suspected, non-selective NSAIDs should be administered with caution.

## Figures and Tables

**Figure 1 vetsci-10-00551-f001:**
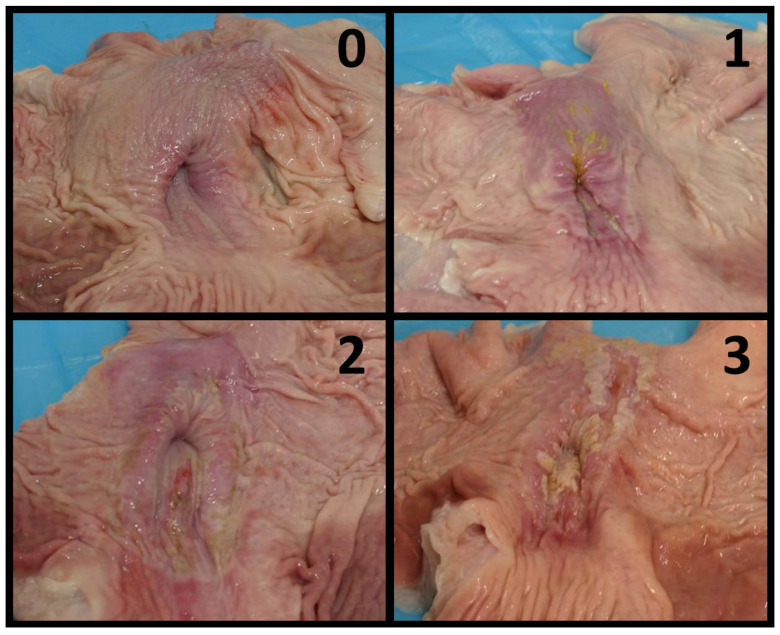
Gastric lesion scoring system applied to the area of interest (*pars oesophagea*). (**0**) No evidence of lesion; (**1**) Hyperkeratosis; (**2**) Erosion and/or mild ulcer; (**3**) Severe ulcer.

**Figure 2 vetsci-10-00551-f002:**
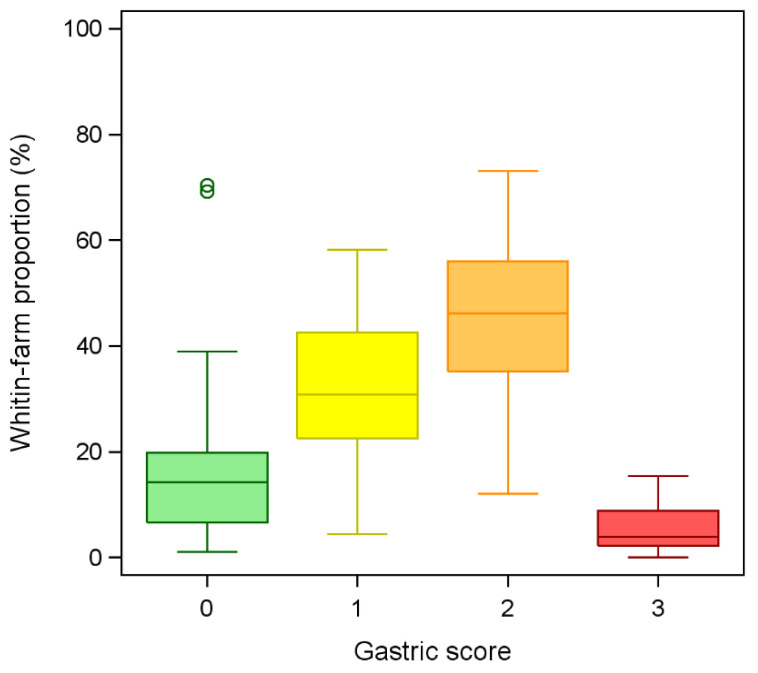
Distribution of the mean proportions of stomachs by gastric ulcer scores observed on Italian pig farms (*n* = 76) following the scoring methodology proposed by Robertson et al. [8] (0 = no lesions; 1 = hyperkeratosis; 2 = erosion/mild ulcer; 3 = severe ulcer).

**Figure 3 vetsci-10-00551-f003:**
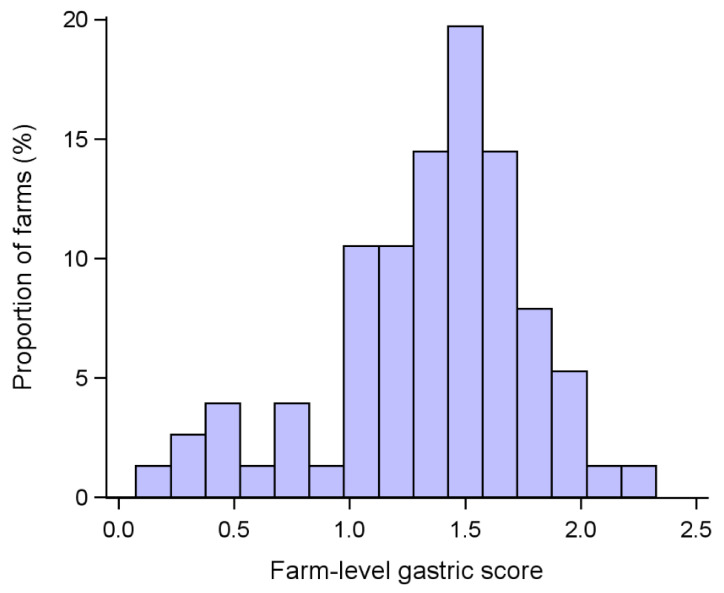
Frequency distribution of farm-level gastric ulcer scores observed on pig farms (*n* = 76).

**Table 1 vetsci-10-00551-t001:** Distribution and descriptive statistics of anti-inflammatory drug use on 36 Italian pig-fattening farms during 2019.

	Overall Use (%)	Farms with Use > 0	Median (Range) TI_1000_ ^†^	Mean ± SD TI_1000_ ^†^
Dexamethasone	11.20	23	0.18 (0–6.17)	0.70 ± 1.31
Nonsteroidal anti-inflammatory drugs	88.80	20	0.07 (0–30.10)	5.22 ± 8.53
*Acetylsalicylic acid*	50.62	10	0 (0–22.07)	2.99 ± 6.50
*Ketoprofen*	0.25	5	0 (0–0.20)	0.02 ± 0.04
*Meloxicam*	0.04	2	0 (0–0.07)	0.003 ± 0.01
*Metamizole*	0.11	3	0 (0–0.13)	0.01 ± 0.03
*Sodium Salicylate*	37.76	5	0 (0–30.01)	2.21 ± 6.62
*Tolfenamic Acid*	0.01	1	0 (0–0.01)	<0.001 ± 0.002
All anti-inflammatory drugs	100	29	0.45 (0–31.6)	5.92 ± 9.04

**^†^** Treatment incidence 1000.

**Table 2 vetsci-10-00551-t002:** Comparative analysis of swine gastric lesion prevalence and severity in reported literature.

Country	Year	No. of Stomachs	Gastric Lesions Score (Prevalence %)				Ref.
			No Lesions	Hyperkeratosis	Erosion/Mild Ulcers	Severe Ulcers	
Italy	2023	9371	20.3	30.7	42.1	6.8	This study
Switzerland	2022	1005	38.8	27.2	14.9	19.1	[41]
Poland	2021	32,264	28.1	9.2	7.8	54.9	[3]
Denmark	2018	447	5.1	24.7	25.1	45.2	[2]
Scotland	2018	78	20.5	46.1	24.4	9.0	[2]
Austria	2018	233	2.6	58.8	10.3	28.3	[42]
Italy	2017	22,551	16.8	62.5	16.6	4.1	[4]
Denmark	2017	712	27.9	33.2	27.3	11.6	[43]
Ghana	2016	75	74.7	18.7	6.76	0.0	[44]
Nigeria	2015	480	67.1	22.5	7.9	2.5	[45]
Italy	2013	635	24.6	22.7	36.3	16.4	[30]
UK	2012	9827	20.4	49.2	23.9	6.4	[5]
Australia	2007	280	3.9	35.0	30.0	31.1	[46]
Switzerland	2005	1897	40.5	44.4	5.0	10.1	[47]
Canada	2002	1021	6.2	42.5	35.8	15.5	[1]
UK	1997	1242	57.0	20.1	9.5	13.4	[33]
USA	1996	80	2.7	39.7	39.7	17.8	[48]

## Data Availability

The data included in this study can be provided upon reasonable request to the corresponding author.

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
