# Peer review of "A Preliminary Study on the Relationship between Gastric Lesions and Anti-Inflammatory Drug Usage in Heavy Pigs"

_vetsci, 2023, doi:10.3390/vetsci10090551_

Round 1

Reviewer 1 Report

The authors presented a well written study on the insurgence of gastric ulcers in heavy pigs in case of use of NSAIDs and steroids. The topic is little explored in literature, and the soundness is quite relevant as the use of anti inflammatory drugs might be linked also to the reduction of antimicrobial use on farm. However, some doubts arose, as detailed in the attached file.

The paper is well written and only minor spelling errors appeared.

Reviewer 2 Report

This study assessed the prevalence of gastric ulcers and their possible association with the use of anti-inflammatory drugs in heavy pigs in Italy, analyzing 9371 pigs from 76 farms. This study has implications for the rational use of anti-inflammatory drugs to improve animal welfare, further confirming the importance of this issue in pig production.

I have only a few minor suggestions for the authors to consider:

1. There is a difference between "gastric ulcers" in the title and "gastric lesions" in the following, so please standardize the terminology.

2. Figure 2: What do the graphs above the "0" group mean?

3. Lines 208-214: The article description does not match Table 1 in some respects.

4. The article does not shed light on the feeding environment of these farms.

5. While the discussion mentions the flaws in the article, the type of feed, diet, etc. for these animals needs to be elucidated.

6. The article examines the relationship between gastric ulcers and the use of anti-inflammatory drugs; is it related to the breed of pig?

7. Were these pigs with gastric ulcers co-infected with other diseases?

8. Some of the references are not formatted consistently, there is no doi, and some of the references are a bit dated.

Round 2

Reviewer 1 Report

The authors answered to the questions.

Minor mistakes have been detected.